# Study and Analysis of a Multi-Mode Power Split Hybrid Transmission

**Xiaojiang Chen [1],\* , Jiajia Jiang [2], Lipeng Zheng [2], Haifeng Tang [2] and Xiaofeng Chen [2]**

[1]  Great Wall Motor Austria Research & Development GmbH, 2542 Kottingbrunn, Austria
[2]  HYCET E-Drive System Co., Ltd., Baoding 071000, China; jiangjiajia@gwm.cn (J.J.);
    zhenglipeng@hycet.com (L.Z.); tanghaifeng@hycet.com (H.T.); chenxiaofeng@hycet.com (X.C.)
\*  Correspondence: xiaojiang.chen@gwm.at; Tel.: +43-660-7048-729

**Abstract:** A two-motor power-split dedicated hybrid transmission (DHT) with two planetary gears is proposed for the applications of a hybrid electric vehicle (HEV) and plug-in HEV (PHEV). The proposed DHT can provide electronically controlled continuous variable transmission (eCVT) with two different gear ratios. One of two electric motors is employed to act as a speeder for splitting the input power of internal combustion engine (ICE) and the other acts as a torquer to assist ICE for boosting. Assisted by an electric motor, ICE can always be enhanced to operate at its efficient area for the benefits of fuel economy improvement. The maximum ICE torque is viable to be mechanically transmitted to vehicle wheels from standstill with two different gear ratios. This feature can help reduce the traction motor torque and power sizing significantly. The paper presents detailed theoretical analyses of the proposed eCVT. Comprehensive simulation demonstrations for a pickup truck HEV application are given to address that the vehicle fuel consumption can be considerably reduced without compromising acceleration performance.

**Keywords:** power split; dedicated hybrid transmission (DHT); planetary gear; electronically controlled continuous variable transmission (eCVT); fuel consumption

## 1. Introduction

Global vehicle manufacturers are demanded to produce more fuel-efficient and low-emission vehicles including electric hybrid vehicles (HEV), plug–in HEV (PHEV) and battery electric vehicles (BEV) in order to satisfy more stringent fuel consumption and emission requirements from global governments. The BEV sale has been growing fast in recent years. However, the further wide acceptance for BEV relies heavily on the battery cost and reliability improvement and battery charging infrastructure development. Comparatively, an HEV or PHEV with a smaller traction battery is not or less dependent on charging facilities, and becomes more viable and market-demanding in a short and medium term.

Due to the significant cost reduction of electric machines and their drive electronics over the years, two-motor based dedicated hybrid transmission (DHT) technologies are becoming a fast growing hybrid powertrain trend for HEV and PHEV applications such as Toyota power-split system hybrid synergy drive (HSD) [1–6], Honda intelligent multi-mode drive (iMMD) [7–9] and two-mode electric variable transmission (EVT) of the General Motor (GM) Voltect-2 [3,10–12]. For a HEV application, the battery size can be minimized to cut the overall hybrid powertrain cost by a dual-motor DHT. An HEV assisted by a single electric motor with a small battery can hardly make a very promising fuel economy improvement over its conventional version model. For a PHEV application, onboard traction batteries have to be sized large enough to satisfy EV driving functions. The popular P2 parallel hybrid powertrain architecture [6,13,14] is adopted by some European vehicle manufacturers for PHEV

applications. However, the P2 parallel hybrid is not fuel efficient and cost-effective enough for wide HEV adoptions because a P2 motor with limited power due to a small HEV traction battery can only achieve a mild hybridization function. A DHT with dual motors can easily provide strong hybrid function without the need of adjusting power and torque rating of two motor drive systems for both HEV and PHEV applications. A single-motor multi-gear hybrid special gearbox is proposed with two planetary gear sets, 2 clutches and 2 brakes to provide the pure electric mode with two gears and the hybrid mode with four gears intended for HEV and PHEV applications [15]. However, its shifting strategy is relatively complex. Its fuel economy improvement will be limited if a small HEV battery is adopted.

A DHT with two motors can have many advantages over a single-motor based parallel hybrid powertrain. Firstly, it can offer smooth and seamless torque transmission between the motors and engine. The overall DHT powertrain cost equipped with a downsized engine can be comparable to a conventional non-hybrid powertrain with a complex multi-geared automatic transmission and turbo-charged engine system due to the significant cost reduction of motor systems in recent years. Furthermore, HEV and PHEV applications can share a common DHT platform without the need of adjusting motor rating. As one of two DHT motors can provide traction assistant power and the other motor can run in generation to compensate power delivery from traction batteries, the traction battery power can be minimized with the significant benefit of prolonging battery life. Additionally, the implementation of the DHT powertrain control strategy is comparatively simple. Moreover, DHT dual motors can offer much more flexibilities for engines to operate in their most efficient region for the benefit of significant fuel economy and emission improvements.

Toyota has produced its one-mode power-split series-parallel hybrid synergy drive (HSD) since 1997 [2–5]. The DHT powertrain architecture of the Toyota's latest HSD with a single planetary gear is shown in Figure 1a [1]. It has a simple mechanical structure and is cost-effective especially for economy-class vehicle applications. The HSD traction torque capability is limited because the mechanical path of engine torque to DHT output only has one fixed gear ratio. Accordingly, a traction motor generator 2 (MG2) with high torque capabilities has to be employed if a high traction application is required. This will unavoidably result in a larger-sized MG2 motor and inverter, and cost increase in the motor system. Furthermore, the MG2 speed will vary linearly with the vehicle speed due to the MG2's direct mechanical coupling with a fixed high gear ratio. During highway cruising, HSD will operate in a direct engine drive mode while the traction motor MG2 has to run persistently at a high speed without delivering any active torque assistance and this would cause extra system loss to the hybrid powertrain. Thereby, it will introduce fuel-consumption penalty and not be efficient enough during continuous highway driving.

Honda launched its first-generation strong hybrid vehicle of a PHEV accord [7–9] in 2014. The simplified architecture of Honda iMMD shown in Figure 1b only employs simple axial transmission gear to achieve eCVT function through a series hybrid mode during city driving and instant highway power boosting. A parallel hybrid mode at high speed is employed by closing the clutch to directly transfer engine torque to the powertrain output shaft with a fixed gear. The maximum traction torque and power of Honda iMMD HEV powertrain fully relies on its traction motor capability. The iMMD traction motor has to be sized to provide full traction torque and power requirements. Moreover, its generator motor must be sized to match completely with its engine rating. The iMMD DHT system will have highest motor sizing requirements compared to Toyota HSD [1–6] and GM Voltec-2 eCVT architecture [10–12]. Hence Honda iMMD HEV eCVT powertrain requires high DHT cost especially for a high-performance HEV application but owns significant benefits in compact transmission mechanical structure and simple powertrain control strategy.

GM introduced its innovative two-mode power-split hybrid powertrain EVT DHT technology based on two planetary gears, as shown in Figure 1c [10–12]. GM dual-mode power-split system can provide 2-geared eCVT functions and result in significant reduction in motor torque and power rating. It is much more scalable for HEV applications with different vehicle platforms. The GM EVT

architecture provides two eCVT hybrid functions with two distinctive gear ratios: an input power spilt at first high gear ratio and a compound power split at second lower gear ratio. Its engine torque can directly transfer through its planetary mechanical path to the vehicle wheel in two gear ratios. Assisted by a power-split motor, the maximum engine torque can be delivered mechanically to the vehicle wheel from vehicle standstill. Due to the engine torque availability with two different gear ratios from standstill, the GM EVT motor power and torque rating to satisfy overall powertrain torque and power demands can be dramatically degraded. This will result in significant overall motor system cost reduction accordingly. Furthermore, its traction motor MG2 in GM EVT can operate as a speeder to split engine torque and power during the compound power-split eCVT. When the vehicle drives at high speed, direct engine drive can be applied by controlling the traction motor MG2 speed around zero speed. Accordingly, two motors in the GM two-mode EVT are not necessarily required to operate at a very high speed during highway driving. This will simply motor control algorithm and flux-wakening control effort for two motors at high speed. As the MG2 loss induced by power-split balancing torque is much lower than that of the MG2 idling loss at high speed, the overall powertrain efficiency of GM two-mode EVT DHT shall be more efficient than Toyota HSD and Honda iMMD DHT during continuous highway cruising. In addition, as motor power and torque requirements are degraded and two motors need not operate to a very high rotational speed, a bidirectional Direct-Current(DC)-to-DC buck-boost converter connecting the onboard traction battery to motor inverters DC-link bus used in Toyota HSD and Honda iMMD is not necessarily required anymore. This will certainly result in the system efficiency improvement and further cost reduction of the GM two-mode EVT DHT powertrain.

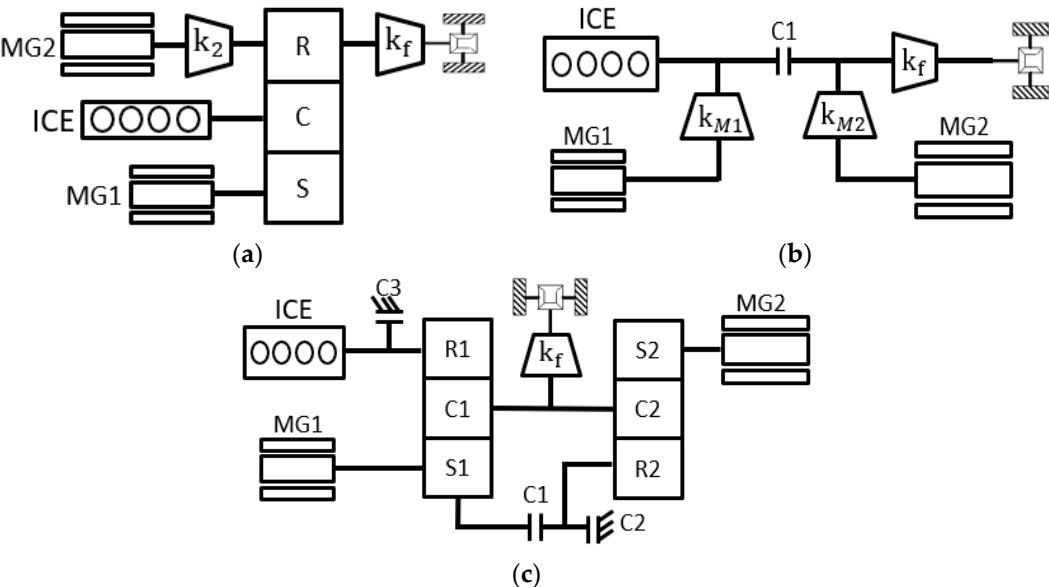

**Figure 1.** Three typical electronically controlled continuous variable transmission (eCVT) dedicated hybrid transmission (DHT) architectures: (**a**) Toyota hybrid synergy drive (HSD); (**b**) Honda intelligent multi-mode drive (iMMD); (**c**) GM two-mode electric variable transmission (EVT).

Ravigneaux planetary gearsets have attracted many interests in DHT applications [16,17]. Configuration syntheses for novel hybrid transmissions consisting of a Ravigneaux gearset and a single planetary gearset are addressed based on graph-theory and lever analogy method [16]. A dual-motor based DHT containing a modified Ravigneaux gearset with a common ringer gear, a common carrier and two sun gears is studied for a PHEV application [17]. Two additional brake clutches are adopted for allowing the hybrid function switching of EV, compound power split and parallel hybrid in a fixed gear [17]. Due to the lack of an input power-split configuration, this Ravigneaux based DHT architecture [17] will fully rely on its EV function provided by two motors to ensure its

maximum traction capability. Once its PHEV battery is depleted, the compound power-split hybrid function can only offer a limited gradeability.

This paper will present a novel concept of two mode eCVT hybrid powertrain architecture that contains less component than that of GM EVT. This eCVT DHT can provide two EV drive modes, two power split modes with two separate gear ratios: an input power split at first gear and a compound power split at second gear, and two direct engine drive or parallel hybrid modes with two same gears. Detailed theoretical analyses are carried out. A comprehensive simulation demonstration for a pickup HEV truck application is finally presented.

## 2. Proposed eCVT Concept

### 2.1. Two-Mode Power Split eCVT Architecture

The proposed two-mode power-split eCVT is based on two planetary gearsets, as shown in Figure 2. The internal combustion engine (ICE) engine torque input is directly connected to the ring gear R1 of first planetary gear PG1. The first motor generator (MG1) is directly coupled to the PG1's sun gear S1. The second motor generator (MG2) is attached to the sun gear S2 of second planetary gear PG2. A clutch CL1 and a brake BK1 are employed between the PG1 ring gear R1 and PG2 carrier C2 for granting different drive functions. The PG1 carrier C1 and PG2 ring gear R2 are mechanically connected permanently to the eCVT output gear transmission that has a fixed gear ratio of $k_f$ to the vehicle wheel output.

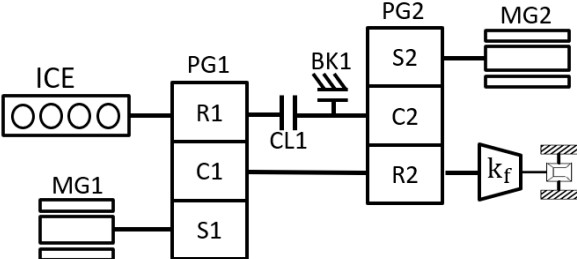

**Figure 2.** Proposed eCVT DHT architecture.

### 2.2. eCVT Drive Mode Introduction

The proposed eCVT in Figure 2 can provide multi-mode drive functions: pure electric-vehicle driving (EV), eCVT hybrid mode and parallel hybrid mode (PH) including direct engine drive, as summarized in the following Table 1.

**Table 1.** Drive modes of proposed electronically controlled continuous variable transmission (eCVT).

| Mode | CL1 | BK1 | ICE | MG1 | MG2 | Definition |
|------|-----|-----|-----|-----|-----|------------|
| EV1 | Open | Close | ○ | ○ | ● | Only MG2 provides pure EV drive |
| EV2 | Close | Close | ○ | ● | ● | Both MG1 and MG2 provide EV drive |
| eCVT1 | Open | Close | ● | ● | ● | Input power split mode at 1st gear |
| PH1 | Open | Close | ● | ● | ● | Parallel hybrid or direct engine drive at 1st gear |
| eCVT2 | Close | Open | ● | ● | ● | Compound power split at 2nd gear |
| PH2 | Close | Open | ● | ● | ● | Parallel hybrid or direct engine drive at 2nd gear |

CL: Clutch; BK: Brake; ICE: Internal Combustion Engine; MG: Motor Generator; EV: Electric Vehicle; eCVT: electronically controlled Continuous Variable Transmission; PH: Parallel Hybrid.

## 3. Theoretical Analyses on Drive Modes

### 3.1. EV1 Mode

The brake BK1 is engaged close and clutch CL1 open during the first EV driving mode – EV1, as shown in the configuration of Figure 3.

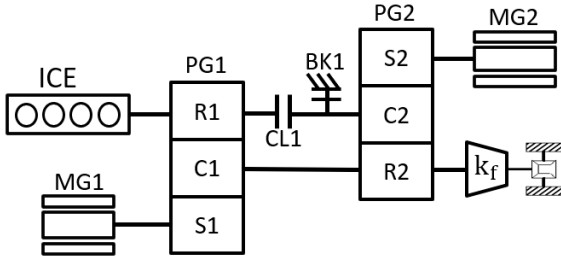

**Figure 3.** eCVT DHT configuration in Electric Vehicle (EV1) mode.

The speed relationship of three power sources can be described as:

$$\omega_{R1-ICE} = 0 \tag{1}$$

$$\omega_{S1\_MG1} = (k_1 + 1)\omega_{C1R2} = (k_1 + 1)k_f\omega_{WHL} \tag{2}$$

$$\omega_{S2\_MG2} = -k_2\omega_{C1R2} = -k_2k_f\omega_{WHL} \tag{3}$$

where $k_1$ and $k_2$ are defined as the gear ratios of the sun gear to ring gear of PG1 and PG2, respectively; $\omega_{R1\_ICE}$ represents the engine angular rotation speed at the PG1 ring gear R1 input shaft, $\omega_{S1\_MG1}$ as the MG1 angular speed at PG1 sun gear S1 input shaft, $\omega_{S2\_MG2}$ as the MG2 angular speed at the PG2 sun gear S2 input shaft and $\omega_{WHL}$ as the final vehicle wheel angular speed.

Torque outputs of three power sources of ICE, MG1 and MG2 are defined by:

$$T_{R1\_ICE} = 0 \tag{4}$$

$$T_{S1\_MG1} = 0 \tag{5}$$

$$T_{WHL} = -k_2k_fT_{S2\_MG2} \tag{6}$$

where $T_{WHL}$ represents the final wheel output torque from the eCVT DHT powertrain. Based on Equations (4)–(6), MG2 provides active drive torque solely to the powertrain with a fixed gear ratio of $-k_2k_f$; due to the engine intrinsic drag torque, engine is kept at standstill without active drive torque delivered to the drivetrain and MG1 will rotate in synchronization with the vehicle speed as described in Equation (2) without active torque delivery.

The maximum vehicle speed in the EV1 mode depends on the maximum allowable rotational speed of the sun gears of two planetary gears PG1 and PG2. The gear ratio of $k_1$ and $k_2$ shall be properly selected to ensure the required maximum EV vehicle speed achievable. EV1 will be used as the prime electric drive mode if EV driving load requests are moderate.

### 3.2. EV2 Mode

When electric drive requires a high traction torque that cannot be solely satisfied by MG2, both the clutch CL1 and brake BK1 shall be closed to allow MG1 and MG2 to provide EV driving simultaneously, as shown in the configuration of Figure 4.

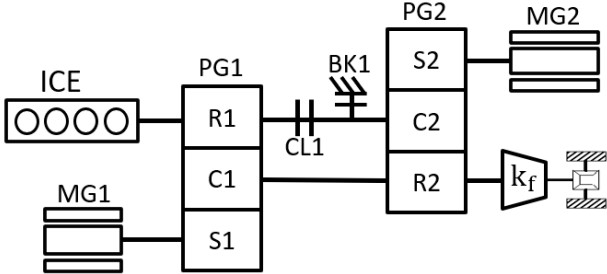

**Figure 4.** eCVT DHT configuration in EV2 mode.

In the EV2 mode, the speed relationship of ICE, MG1 and MG2 still complies with Equations (1)–(3). However, the wheel drive torque shall be provided by both MG1 and MG2 as described by Equations (7) and (8):

$$T_{R1C2\_ICE} = 0 \tag{7}$$

$$T_{WHL} = (k_1 + 1)k_f T_{S1\_MG1} - k_2 k_f T_{S2\_MG2} \tag{8}$$

MG1 and MG2 work jointly to provide traction power and torque together to achieve electric drive function. This will help to alleviate the thermal stress when the vehicle is driven by one motor alone in a high load condition. Optimal torque split between two motors will help improve the EV drive efficiency and gradeability. In addition, the overall maximum EV power and torque rating to two motors can be split. This will obviously result in smaller motor sizing requirements for cost-down benefits.

### 3.3. eCVT1 Mode—Input Power Split

The eCVT1 mode has the same configuration as EV1 as illustrated in Figure 5.

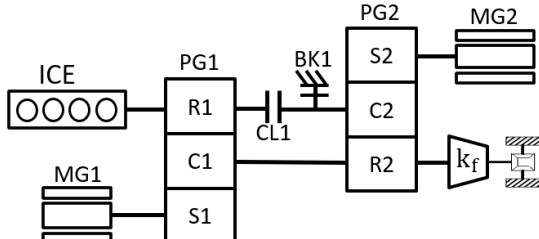

**Figure 5.** eCVT DHT configuration in eCVT1 and parallel hybrid (PH)1 mode.

In this input power-split eCVT mode [6], the MG1 speed shall be controlled in a closed speed-feedback loop to maintain optimal ICE engine running points. MG1 automatically provides a split torque to balance with the engine torque input. In the meantime, MG2 generates extra torque occasionally to assist the powertrain traction or balance the power of onboard traction battery. In such an input power split case, the speed relationship of three power sources is described by the following Equations (9) and (10):

$$\omega_{S1\_MG1} + k_1 \omega_{R1\_ICE} = (k_1 + 1)\omega_{C1R2} = (k_1 + 1)k_f \omega_{WHL} \tag{9}$$

$$\omega_{S2\_MG2} = -k_2 \omega_{C1R2} = -k_2 k_f \omega_{WHL} \tag{10}$$

The engine input power is split continuously by the speeder MG1 into two separate power transmission path: mechanical path and electro-mechanical path. For the electro-mechanical path, the speeder MG1 will provide a split torque to balance the internal force distribution of the first planetary gear PG1. In a steady state by neglecting some minor disturbance effects, the MG1 split torque can ideally be described in Equation (11):

$$T_{S1\_MG1} = \frac{1}{k_1} T_{R1\_ICE} \tag{11}$$

The MG1 split power to the overall ICE power input is further derived by:

$$P_{S1\_MG1} = T_{S1\_MG1}\omega_{S1\_MG1} = \frac{1}{k_1} T_{R1\_ICE}\omega_{S1\_MG1} \tag{12}$$

The engine input torque of $T_{R1\_ICE}$ has to be always positive, then the MG1 split power $P_{S1\_MG1} < 0$ when $\omega_{S1\_MG1} < 0$, i.e., MG1 generative power is provided to either charge battery or directly to the MG2 system; $P_{S1\_MG1} = 0$ when $\omega_{S1\_MG1} = 0$, i.e., no active ICE power is split; $P_{S1\_MG1} > 0$ when

$\omega_{S1\_MG1} > 0$, i.e., MG1 consumes the battery power to provide extra motoring power to be added with ICE input power to the final eCVT power output.

Furthermore, the eCVT DHT powertrain output torque will contain two components provided by ICE and MG2 separately as descried in Equation (13):

$$T_{WHL} = k_f T_{C1R2} = \left( \frac{k_1 + 1}{k_1} T_{R1\_ICE} - k_2 T_{S2\_MG2} \right) k_f \tag{13}$$

ICE will provide a mechanical torque directly to the output shaft of the PG1 carrier C1 with a fixed gear ratio of $\frac{k_1+1}{k_1}$ regardless of the vehicle speed. The engine mechanical power output to the wheel through the planetary mechanical path depends on Equation (14):

$$P_{ICE\_MO} = \frac{k_1 + 1}{k_1} T_{R1\_ICE} \omega_{C1R2} = \frac{k_1 + 1}{k_1} k_f T_{R1\_ICE} \omega_{WHL} \tag{14}$$

The power split ratio of ICE mechanical output power $P_{ICE\_MO}$ versus ICE overall input power at PG1 ring gear R1 is derived by:

$$\beta_{eCVT1} = \frac{P_{ICE\_MO}}{T_{R1\_ICE} \omega_{R1\_ICE}} = \frac{k_1 + 1}{k_1} \frac{\omega_{C1R2}}{\omega_{R1\_ICE}} = \frac{k_1 + 1}{k_1} \frac{1}{\varphi_{eCVT1}} \tag{15}$$

where $\varphi_{eCVT1} = \frac{\omega_{R1\_ICE}}{\omega_{C1R2}}$ is defined as the ICE eCVT1 gear ratio to the PG1 output, which will be continuously variable with the vehicle speed and driving load demands. At zero vehicle speed, the eCVT1 ICE gear ratio $\varphi_{eCVT1}$ is infinite with ICE power split ratio $\beta_{eCVT1} = 0$. This implies that the ICE input power is completely converted into electricity by MG1 through the electro-mechanical path ideally. When the power split ratio $\beta_{eCVT1} = 1$, i.e., $\varphi_{eCVT1} = \frac{k_1+1}{k_1}$, the MG1 speed must be zero, i.e., $\omega_{S1\_MG1} = 0$. In this special case, the ICE input power will be completely transmitted to the vehicle wheel through the planetary mechanical path ideally. The principle of eCVT1 power split can be clarified by the planetary-gear lever diagram as illustrated in Figure 6.

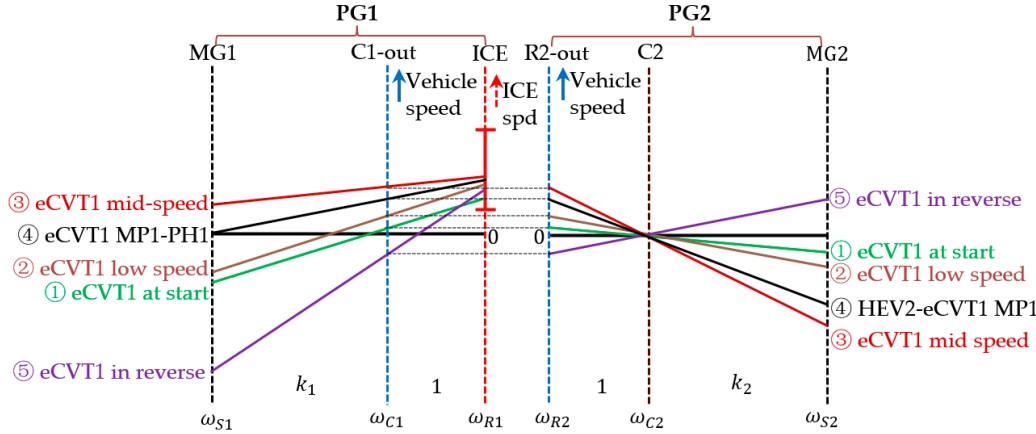

**Figure 6.** Lever schematics for eCVT1 and first parallel hybrid PH1 mode.

Figure 6 illustrate that MG2 will always operate at a negative rotational speed when the vehicle drives forwards in the eCVT1 mode because of the speed relationship described in Equation (10). In the meantime, the ICE speed at the PG1 ring gear input shaft shall be continuously and optimally adjusted by MG1 through a feedback speed control. Firstly, an expected ICE speed shall be calculated in real time based on a lookup map versus the inputs of the drive load and vehicle speed, or by an online optimization algorithm. Furthermore, the MG1 speed reference shall be calculated based on Equation (9), where the current vehicle wheel speed and expected ICE speed are employed as

inputs. Eventually MG1 is commanded in a closed-loop control to follow its speed reference in order to maintain the required optimal engine speed.

As indicated in the operation Line 1–4 of Figure 6, MG1 will run at a negative speed from startup in generation mode to split the ICE power and gradually move towards zero speed with the increase of the vehicle speed. At zero speed, MG1 has zero power split but has to maintain its split torque described in Equation (11) to allow ICE input power directly be transferred to the wheel completely; once MG1 has to run into a positive speed, MG1 starts to consume battery energy in motoring mode to deliver extra positive power adding to existing ICE input power at the powertrain output. If the onboard battery power has to be balanced, MG2 must provide generative power to counteract the MG1 motoring power. In such a case, power re-circulation will happen inside of the planetary system and result in the decrease of eCVT transmission efficiency. With the further increase of MG1 speed at position rotation, overall eCVT efficiency will be even less. Thus, the eCVT1 power split shall be avoided to be applied at high vehicle speed. The eCVT1 operational range of vehicle speed depends on both the restriction of the maximum allowable speed of the PG2 sun gear and overall powertrain efficiency consideration.

Based on Equation (13), the engine torque to the final wheel output through planetary gear mechanical pass has a fixed gear ratio even though the engine speed to the final output continuously varies:

$$k_{G1\_eCVT1} = \frac{k_1 + 1}{k_1} k_f \tag{16}$$

Then eCVT1 is defined as the eCVT mode at the first gear of $k_{G1\_eCVT1}$.

### 3.4. PH1 Mode

The first parallel hybrid (PH1) mode is defined to be the special operational case within the eCVT1 mode when MG1 is controlled at zero speed, as indicated in Line–4 of Figure 6. Based on Equations (9), (10) and(15), we can derive:

$$\omega_{S1\_MG1} = 0 \tag{17}$$

$$\varphi_{eCVT1\_PH1} = \frac{\omega_{R1\_ICE}}{\omega_{C1R2}} = \frac{k_1 + 1}{k_1} \tag{18}$$

$$\beta_{eCVT1\_PH1} = \frac{P_{ICE\_MO}}{T_{R1\_ICE}\omega_{R1\_ICE}} = \frac{k_1 + 1}{k_1} \frac{\omega_{C1R2}}{\omega_{R1\_ICE}} = 1 \tag{19}$$

At this operational point in the eCVT1 mode, the ICE power split ratio $\beta_{eCVT1\_PH1} = 1$ and the MG1 active split power $P_{S1\_MG1} = 0$. This implies the whole ICE input power is delivered to the vehicle wheel through the planetary mechanical path if eCVT system component losses are ignored. This special operation point is defined as the first Mechanical Point (MP1) of eCVT [11,12,18]. The torque transmission through the planetary-gear mechanical path is generally more efficient than through electro-mechanical path. Hence this special operation node is employed to implement direct engine drive or parallel hybrid mode of ICE and MG2 at the first gear.

In the PH1 mode, the torque relationship of three power sources are the same as described in Equation (11) and Equation (13). Even though the split device MG1 is controlled at zero speed with zero active split power ideally in the PH1 mode, MG1 still has to provide a balanced split torque against ICE torque as defined in Equation (11). The MG1 steady-state split torque will unavoidably cause extra losses including motor winding copper loss and motor inverter loss. In order to balance the thermal stress of MG1 motor stator windings and inverter phases, it is preferable to operate MG1 in a low rotational speed during the PH1 mode.

### 3.5. eCVT2 Mode—Compound Power Split eCVT

The eCVT2 mode has a configuration as illustrated in Figure 7. The clutch CL1 is engaged close and the brake BK1 is kept open.

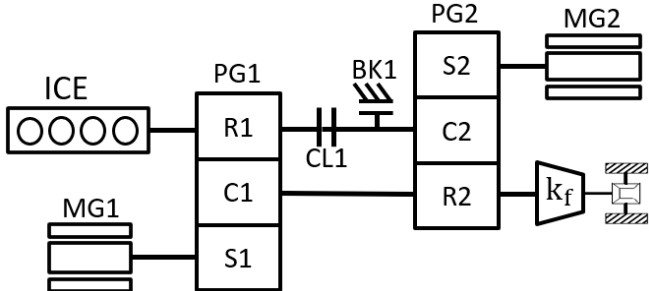

**Figure 7.** eCVT DHT configuration in eCVT2 and PH2 mode.

eCVT2 is a compound power-split hybrid mode [6,10–12,18]. Two planetary gearsets of PG1 and PG2 are restructured into a compound planetary system which allows either MG1 or MG2 to act as the speeder for splitting the ICE input power and the other as the torquer. The basic speed relationship of three power sources in this compound system can be further derived as:

$$\omega_{S1\_MG1} + k_1\omega_{R1C2\_ICE} = (k_1 + 1)\omega_{C1R2} \tag{20}$$

$$\omega_{S2\_MG2} + k_2\omega_{C1R2} = (k_2 + 1)\omega_{R1C2\_ICE} \tag{21}$$

$$\omega_{C1R2} = k_f\omega_{WHL} \tag{22}$$

where $\omega_{R1C2\_ICE}$ represents the ICE engine speed at the joint shaft of PG1 ring gear R1 and PG2 carrier C2. The speed relationship and steady-state torque relationship of ICE, MG1, MG2 and eCVT output can be illustrated by a reconstructed lever diagram as shown in Figure 8.

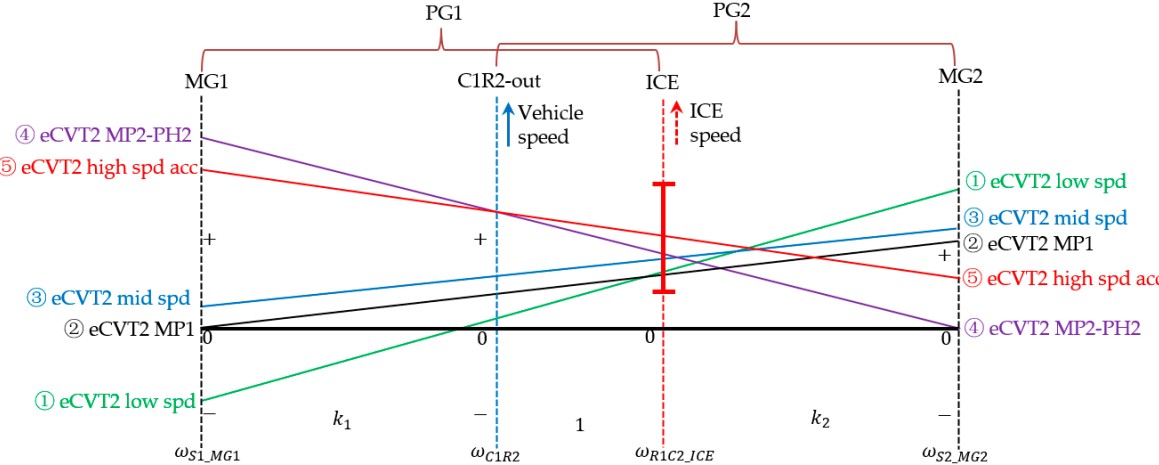

**Figure 8.** Lever schematics of compound planetary system in eCVT2 and PH2 mode.

As indicated in Figure 8, the speeds of ICE, MG1, MG2 and the output shaft of the compound planetary system respectively position along the same lever. The planetary-gearset output speed $\omega_{C1R2}$ is linear to the vehicle wheel speed with a fixed gear ratio of $k_f$. Two motor speeds of $\omega_{S1\_MG1}$ and $\omega_{S2\_MG2}$, respectively located at two sides of $\omega_{C1R2}$. The ICE speed $\omega_{R1C2\_ICE}$ is along the same side of $\omega_{S2\_MG2}$.

Once one speed of three-power sources is determined along with the known vehicle speed, the speed lever line is fixed accordingly in Figure 8. ICE is preferably employed as main active power source in torque control mode to meet the driver request. The optimal running point of ICE speed can be continuously and variably adjusted by active closed-loop speed control of either MG1 or MG2. The resistive torque of the speed adjuster of either MG1 or MG2 will automatically respond and generate a balanced torque against the ICE applied torque load within the compound lever. The other

motor can occasionally provide instantaneous assistant torque to ICE or regenerative braking torque for mechanical energy recovery during vehicle braking operation.

Based on the lever schematics illustrated in Figure 8, this compound power split eCVT can operate in the whole vehicle speed range without violating the thresholds of maximum speed limit of sun gears of PG1 and PG2. In this compound planetary lever, either MG1 or MG 2 can be employed as the power-split device in a closed speed-loop control to optimally adjust the required engine speed, however the eCVT transmission performance under either MG1 or MG2 as the speed variator will behave differently.

In order to derive the torque relationship of three power sources of ICE, MG1 and MG2 of the reconstructed compound planetary system, speed lever relationships in Equations (20) and (21) are further reorganized into two different forms as discussed in the following sections.

### 3.5.1. Steady-State eCVT2 Torque Distribution with Speeder MG1

Equations (20)–(22) is reorganized into the form of Equations (23)–(25) if MG1 is adopted as the power-split speeder to ICE input:

$$\omega_{S1\_MG1} + k_1\omega_{R1C2\_ICE} = (k_1 + 1)\omega_{C1R2} \tag{23}$$

$$k_1\omega_{S2\_MG2} + (k_2 + 1)\omega_{S1\_MG1} = (k_1 + k_2 + 1)\omega_{C1R2} \tag{24}$$

$$\omega_{C1R2} = k_f\omega_{WHL} \tag{25}$$

By carefully examining the lever schematics in Figure 8 and speed relationship of Equations (23)–(25), and disregarding the planetary-gear transmission loss and some negligible dynamics coupling factor [6], the speed $\omega_{C1R2}$ is used as the imaginary lever fulcrum, then the resistive steady-state torque response of the MG1 speed loop output will have to contain two terms from both ICE and MG2 applied torque, respectively as described by Equation (26):

$$T_{S1\_MG1} = \frac{1}{k_1}T_{R1C2\_ICE} + \frac{k_2 + 1}{k_1}T_{S2\_MG2} \tag{26}$$

In order to derive the final torque output of this compound eCVT transmission, the speeder MG1 shall be used as the imaginary lever fulcrum. The eCVT transmission torque output to the wheel can be derived as in Equation (27) containing two terms from both ICE and MG2 applied torque, respectively [18]:

$$T_{WHL} = k_fT_{C1R2} = \frac{k_1 + 1}{k_1}k_fT_{R1C2\_ICE} + \frac{k_1 + k_2 + 1}{k_1}k_fT_{S2\_MG2} \tag{27}$$

As we can see, if MG1 is used as the power-split device in the eCVT2 mode, the final gear ratio of ICE torque to the wheel in Equation (27) is completely same as the first gear ratio of the eCVT1 mode described in Equation (16).

$$k_{G1\_eCVT2} = k_f\frac{k_1 + 1}{k_1} = k_{G1\_eCVT1} \tag{28}$$

Furthermore, the MG1 split torque described in Equation (26) has the same sign as both ICE and MG2 torque. Consequently, MG1 can only provide generative split power to ICE input power when MG1 run into negative rotational speed when the vehicle speed is low. In the meantime, the positive assistant torque from MG2 will intensify the MG1 split torque and power. This will unavoidably cause the electric power recirculation and eCVT efficiency drop. Furthermore, higher torque and power rating to MG1 has to be required as well. Thus, the eCVT2 control mode by using MG1 as the power split device is not preferable operation mode for a high load drive request. Instead eCVT1 is more efficient in response to high load condition at low vehicle speed.

When the MG1 speed is maintained at standstill, as indicated in the operation line-2 of Figure 8, the MG1 split power is equal to zero. However, MG1 still have to bear the split torque as described in Equation (26) at zero speed. In such a special case, if the loss due to the MG1 balancing torque described in Equation (26) is negligible, ideally the engine and MG2 power will completely transfer to the final wheel. This can be approximately regarded as a parallel hybrid mode of ICE and MG2 torque with fixed gear ratios in Equation (28), or direct ICE drive if no MG2 torque is applied. This running point is defined as the first Mechanical Point (MP1) of eCVT2 mode with a fixed gain of $k_f \frac{k_1+1}{k_1}$. The eCVT2 MP1 is completely same as the eCVT1 MP1. MP1 can be used as the smooth switching point between eCVT1 and eCVT2. The speed lever line across MP1 shown in Figure 8 is not unique but infinite and will vary with ICE speed and vehicle speed.

### 3.5.2. Steady-State eCVT2 Torque Distribution with Speeder MG2

Equations (20)–(22) can also be reconstructed alternatively by Equations (29)–(31) below if MG2 is adopted as the power-split speeder to the ICE input:

$$\omega_{S2\_MG2} + k_2 \omega_{C1R2} = (k_2 + 1)\omega_{R1C2\_ICE} \tag{29}$$

$$k_1 \omega_{S2\_MG2} + (k_2 + 1)\omega_{S1\_MG1} = (k_1 + k_2 + 1)\omega_{C1R2} \tag{30}$$

$$\omega_{C1R2} = k_f \omega_{WHL} \tag{31}$$

Based on the lever principle and the speed $\omega_{C1R2}$ as the imaginary lever fulcrum, assuming the planetary-gear transmission loss and some negligible dynamics coupling factors [6,18] at steady state are negligible, then the resistive torque generated from the MG2 closed-loop speed control will have to contain two terms from both ICE and MG1 applied torque, respectively:

$$T_{S2\_MG2} = -\frac{1}{k_2 + 1} T_{R1C2\_ICE} + \frac{k_1}{k_2 + 1} T_{S1\_MG1} \tag{32}$$

The final torque output of this eCVT transmission can be simply derived in Equation (33) if the speeder MG2 is regarded as the imaginary lever fulcrum.

$$T_{WHL} = k_f T_{C1R2} = \frac{k_2}{k_2 + 1} k_f T_{R1C2\_ICE} + \frac{k_1 + k_2 + 1}{k_2 + 1} k_f T_{S1\_MG1} \tag{33}$$

Equation (33) indicates that the final gear ratio of ICE torque to the wheel has a completely different gain of $\frac{k_2}{k_2+1} k_f$ compared to the first gear ratio of $\frac{k_1+1}{k_1} k_f$ when MG2 is used as the power-split device in the eCVT2 mode. This gear ratio is defined as the second gear ratio for engine torque transmission to the final wheel output.

$$k_{G2\_eCVT2} = \frac{k_2}{k_2 + 1} k_f \tag{34}$$

Based on Equation (33), the MG1 positive torque regardless of the MG1 rotation speed direction will provide acceleration assistance to the engine torque. In the meantime, the MG1 positive torque will accordingly reduce the overall MG2 split torque as described in Equation (32). This results in less MG1 overall split power and can improve the eCVT transmission efficiency. Once the MG1 applied torque has the relationship of $T_{S1\_MG1} = \frac{1}{k_2+1} T_{R1C2\_ICE}$, the MG2 overall split torque $T_{S2-MG2} = 0$, i.e., MG2 split power is zero. This means all input power from ICE and MG1 will fully transfer to the vehicle wheel ideally.

Furthermore, the ICE power split ratio in eCVT2 by MG2 can be further defined as:

$$\beta_{eCVT2\_ICE} = \frac{\frac{k_2}{k_2+1} T_{R1C2\_ICE}\omega_{C1R2}}{T_{R1C2\_ICE}\omega_{R1C2\_ICE}} = \frac{k_2}{k_2+1} \frac{\omega_{C1R2}}{\omega_{R1C2\_ICE}} = \frac{k_2}{k_2+1} \frac{1}{\varphi_{eCVT2\_ICE}} \tag{35}$$

where $\varphi_{eCVT2\_ICE} = \frac{\omega_{R1C2\_ICE}}{\omega_{C1R2}}$ is defined as the ICE eCVT2 speed ratio and will vary continuously with the vehicle speed and driving load requests.

### 3.6. PH2 Mode

When the MG2 speed is controlled to zero speed, the overall MG2 split power is equal to zero. This implies that the ICE input power will be completely transmitted to the vehicle wheel through the planetary mechanical path ideally. This special eCVT2 operation mode is named as the PH2 mode as indicated in the operational line-4 of Figure 8. This is defined as the eCVT2 second Mechanical Point (MP2) [10–12]. Based on Equation (35), we can derive:

$$\omega_{S2\_MG2} = 0 \tag{36}$$

$$\varphi_{eCVT2\_PH2} = \frac{\omega_{R1\_ICE}}{\omega_{C1R2}} = \frac{k_2}{k_2 + 1} \tag{37}$$

$$\beta_{eCVT2\_PH2} = \frac{\frac{k_2}{k_2+1}T_{R1C2\_ICE}\omega_{C1R2}}{T_{R1C2\_ICE}\omega_{R1C2\_ICE}} = \frac{k_2}{k_2+1}\frac{\omega_{C1R2}}{\omega_{R1C2\_ICE}} = 1 \tag{38}$$

In the PH2 mode, the torque dependency of three power sources is same as the equations described in Equations (32) and (33). Even though the MG2 split power split is zero, MG2 must provide a balanced split torque against ICE and MG1 torque input as defined in Equation (32). This MG2 steady-state split torque will unavoidably cause some losses due to motor winding copper loss and motor inverter loss. In a practical application, it is favorable for MG2 to rotate at a low rotational speed when the PH2 mode is employed in order to alleviate the thermal stress unbalance of MG2 motor stator winding and inverter phases.

When the vehicle is driven at a high speed, such as steady cruising control, if the engine can directly operate into its most efficient area, the PH2 mode shall be used as the primary optimal control mode to achieve direct ICE drive. MG1 only provides assistant torque in parallel occasionally to satisfy the instantaneous variable torque requests and in the meantime ICE is commanded to satisfy stable and slow-varying torque requests.

## 4. eCVT Component Sizing Principles

The torque and power rating of MG1, MG2 and ICE shall be sized to meet the vehicle performance requirements including acceleration, gradeability, maximum sustainable vehicle speed, etc. An ideal powertrain performance requirement is illustrated in Figure 9. At low vehicle speed, constant high traction torque is demanded for satisfying acceleration and gradeability requirements. A constant power is needed above a base speed. The torque-speed curve indicated by inner solid line is assumed to be fully provided by the maximum ICE power capability and the extra power and torque illustrated in the outer dotted line is provided by either MG2 or MG1 to consume onboard traction battery power.

As illustrated in Figure 9, in the eCVT1 mode, through the MG1 torque split, ICE maximum torque can constantly be transmitted to the wheel by the first gear ratio of $k_{G1} = \frac{k_1+1}{k_1}k_f$ from zero vehicle speed theoretically. Similarly, the ICE maximum torque can also be provided to the wheel by the second gear ratio of $k_{G2} = \frac{k_1}{k_2+1}k_f$ from zero vehicle speed to high speed in the eCVT2 mode. Therefore, the maximum eCVT powertrain torque output can be derived from Equation (13) as:

$$T_{WHL\_max} = \frac{k_1+1}{k_1}k_f T_{ICE\_max} + k_2 k_f T_{MG2\_max} \tag{39}$$

where $T_{WHL\_max}$ represents the maximum wheel torque demand of eCVT hybrid powertrain output in order to meet the vehicle performance requirements; $T_{ICE\_max}$ is the maximum available ICE torque

output; $T_{MG2\_max}$ is the maximum MG2 positive torque capability. Based on Equation (39), the MG2 maximum torque rating can be sized by:

$$T_{MG2max} = \frac{1}{k_2 k_f} T_{WHL\_max} - \frac{k_1 + 1}{k_1 k_2} T_{ICE\_max} \tag{40}$$

As we can see from Equation (40), due to the support of ICE torque through its mechanical path with the first-gear ratio, the maximum torque and power demand by MG2 can be dramatically reduced. MG2 is only required to provide torque and power to cover the area between the lines of the maximum ICE mechanical torque output at first gear and maximum required wheel torque, as illustrated in Figure 9. If $P_{WHL\_max}$ is defined to be the maximum required powertrain power to the wheel, then the MG2 maximum power demand can be approximately derived by Equation (41):

$$P_{MG2\_max} = k_2 k_f T_{MG2\_max} \frac{P_{WHL\_max}}{T_{WHL\_max}} \tag{41}$$

Based on the torque split principle in Equation (11), the MG1 maximum torque capability can be rated by:

$$T_{MG1\_max} = \frac{1}{k_1} T_{ICE\_max} \tag{42}$$

Accordingly, in the eCVT2 mode, the maximum powertrain torque output to the wheel can be derived based on Equation (33):

$$T_{WHL\_eCVT2\_max} = k_f \left( \frac{k_2}{k_2 + 1} + \frac{k_1 + k_2 + 1}{k_2 + 1} \frac{1}{k_1} \right) T_{ICE\_max} = \frac{k_1 + 1}{k_1} k_f T_{ICE\_max} \tag{43}$$

As we can see from Equation (43), the maximum available torque in the eCVT2 mode is theoretically equal to the maximum ICE mechanical torque output in the eCVT1 mode. In theory, this implied that eCVT2 can fully cover the powertrain torque control area below the $\frac{k_1+1}{k_1} k_f T_{ICE\_max}$ line as illustrated in Figure 9.

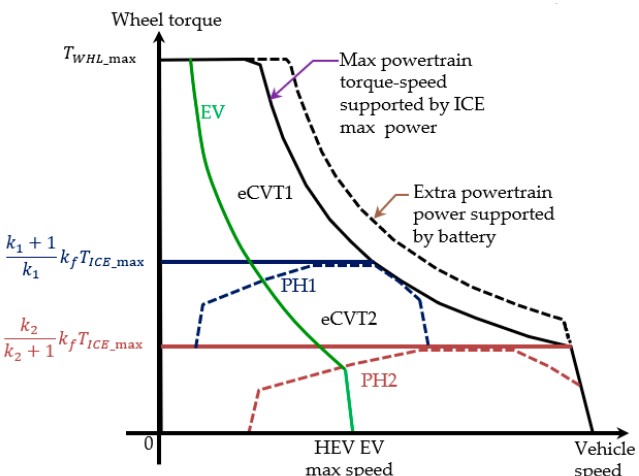

**Figure 9.** eCVT powertrain overall torque–speed performance demand.

Furthermore, based on Figure 9, MG1 shall be able to provide torque and power coverage between the toque output line of $\frac{k_1+1}{k_1} k_f T_{ICE-max}$ and $\frac{k_2}{k_2+1} k_f T_{ICE-max}$ at least. This can be verified by:

$$
\begin{aligned}
T_{WHL\_MG1\_eCVT2\_max} &= \frac{k_1+1}{k_1} k_f T_{ICE\_max} - \frac{k_2}{k_2+1} k_f T_{ICE\_max} \\
&= \frac{k_1+k_2+1}{k_2+1} \frac{k_f}{k_1} T_{ICE\_max} = \frac{k_1+k_2+1}{k_2+1} k_f T_{MG1\_max}
\end{aligned} \tag{44}
$$

This fully complies with the MG1 eCVT2 torque output defined in Equation (33). In addition, the maximum power demand from MG1 can be estimated approximately by Equation (45):

$$P_{MG1\_max} \approx \frac{k_1 + k_2 + 1}{k_2 + 1} k_f T_{MG1\_max} \frac{P_{WHL\_max}}{\frac{k_1+1}{k_1} k_f T_{ICE\_max}} = \frac{k_1 + k_2 + 1}{(k_2 + 1)(k_1 + 1)} P_{WHL\_max} \tag{45}$$

This MG1 power rating definition in Equation (45) can satisfy the maximum power split requirement in the eCVT1 mode as well.

## 5. Simulation Demonstration

### 5.1. Simulation Parameter Setup

A simulation demonstration of the proposed eCVT hybrid powertrain is carried out for a pickup truck HEV application. The relevant parameters of the pickup truck vehicle are listed in Table 2.

**Table 2.** Vehicle parameters of a hybrid electric vehicle (HEV) pickup truck.

| Vehicle Parameters | Definition |
| --- | --- |
| Half-load vehicle weight in simulation | 2357 kg |
| Maximum speed | >160 km/h |
| EV max speed | >105 km/h |
| 0–100 km/h acceleration | <12 s |
| Maximum gradeability | >35% |
| Maximum wheel power | >120 kW |
| Maximum total wheel drive torque | > 3600 Nm |
| Fuel consumption in WLTC [1] | <7.5 L/100 km |
| Tyre radius | 0.376 mm |
| Maximum wheel speed | 1130 rpm@160 km/h |
| Rolling resistance coefficient | 0.0075 |
| Vehicle front area | 2.96 m$^2$ |
| Air resistance coefficient | 0.42 |

[1] WLTC—Worldwide harmonized Light vehicles Test Cycle.

A longitudinal two-geared eCVT DHT transmission is proposed with the schematic mechanical structure layout as shown in Figure 10. The output of the eCVT DHT is directly connected to the final drive axle with a fixed final gear ratio of $k_f = 4.1$. The ICE input shaft is coupled to the eCVT DHT through a torsional damper.

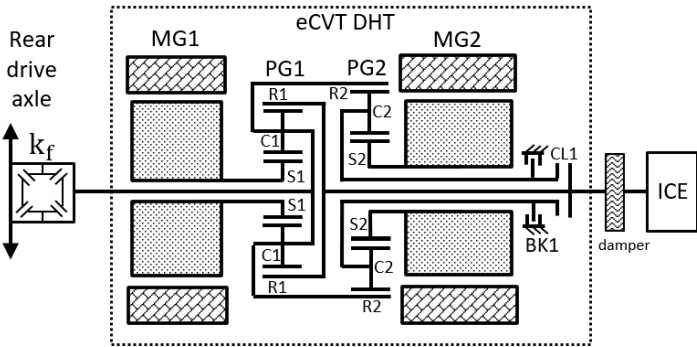

**Figure 10.** Schematic structure layout of the longitudinal eCVT DHT for a pickup truck HEV.

The eCVT DHT component parameters are listed in Table 3. The HEV pickup truck is equipped with a 1.5-L turbo-charged engine with a peak torque-power rating of 264 Nm/110 kW. The gear ratios of two planetary gears are carefully selected to ensure the maximum operational rotation speed of PG1 and PG2 sun gears be restricted within 10,000 rpm. The two gear ratios of eCVT DHT to the final

wheel are selected with $k_{G1} = 6.235$ and $k_{G2} = 2.961$. Two electric machines of MG1 and MG2 are sized based on the theoretical principles addressed in Section 4. The continuous rating of MG1 and MG2 has significant impact on the vehicle continuous gradeability. A high-voltage (HV) traction battery is sized based on an HEV power-type battery cell with a capacity of 5.5 Ah and an instantaneous peak discharge rate of 30 C. The internal resistance of battery cells is modeled based on a lookup table versus the battery state of charge (SOC) data.

**Table 3.** Component parameters of eCVT HEV powertrain.

| HEV Powertrain Component Parameters | | Values |
|---|---|---|
| **ICE** | Displacement<br>Peak torque power<br>Peak efficiency | 1.5-L turbo-charged gasoline engine<br>264 Nm@3600 rpm/110 kW<br>>36% |
| **MG1** | Peak value<br>Nominal value | 140 Nm/65 kW/10,000 rpm<br>80 Nm/35 kW |
| **MG2** | Peak value<br>Nominal value | 220 Nm/70 kW/10,000 rpm<br>120 Nm/40 kW |
| **Gear** | PG1 ratio<br>PG2 ratio<br>Average eCVT mechanical efficiency<br>Final gear ratio of rear-drive axle<br>1st gear ratio (MP1 gain)<br>2nd gear ratio (MP2 gain) | $k_1 = 1.92$<br>$k_2 = 2.60$<br>95%<br>$k_f = 4.10$<br>$k_{G1} = 6.235$<br>$k_{G2} = 2.961$ |
| **HV battery** | Cell<br>Pack capacity | 5.5 Ah, 96 in series, 30 C instantaneous max-discharge rate<br>350 V/1.92 kWh, <60 kW (10 s) |

ICE, MG1 and MG2 are respectively modeled based on their torque-speed-efficiency performance maps as shown in Figure 11. The maximum engine efficiency is around 36.5%. The MG1 and MG2 motor systems including motor and inverter has a peak efficiency of around 94%. The eCVT control strategy is developed to ensure ICE operate ideally along its optimal performance curve as indicated in the ICE efficiency map of Figure 11 for improving fuel economy.

Some important eCVT DHT characteristic parameters are calculated based on the component sizing specifications as listed in Table 4. The maximum speed of PG1 and PG2's sun gears is clamped within 10,000 rpm. The peak torque in both EV and eCVT mode will satisfy the wheel torque requirements in Table 2.

**Table 4.** eCVT dedicated hybrid transmission (DHT) parameters for an HEV pickup truck application.

| HEV Powertrain Component Parameters | | Values |
|---|---|---|
| **Motor maximum speed** | EV MG1 maximum speed<br>EV MG2 maximum speed | ≈8870 rpm @ 105 km/h<br>≈7900 rpm@105 km/h |
| Maximum wheel torque capability | EV maximum wheel torque<br>eCVT1 maximum wheel torque<br>ICE max wheel output torque at 1st gear<br>ICE max wheel output torque at 2nd gear | ≈3820 Nm<br>≈3790 Nm<br>≈1560 Nm@$k_{G1}$<br>≈740 Nm@$k_{G2}$ |
| PH2 max speed | ICE maximum speed<br>MG1 maximum speed | ≈3340 rpm@160 km/h<br>≈7100 rpm@160 km/h |

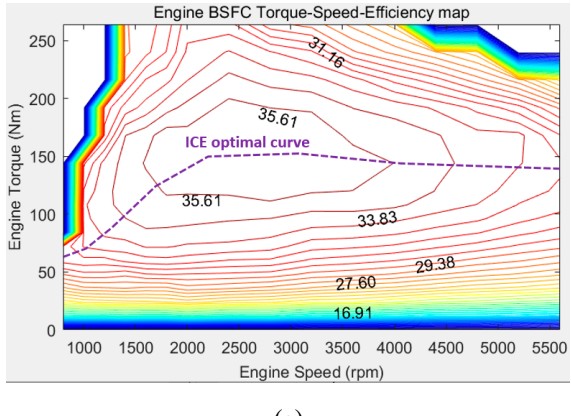

(a)

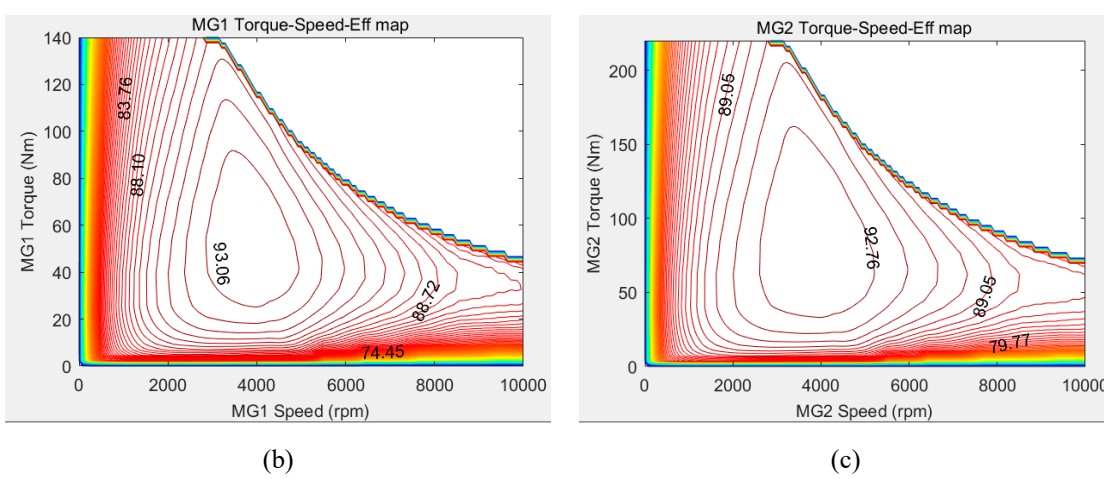

(b)                                                            (c)

**Figure 11.** ICE, motor generator (MG)1 and MG2 torque-speed-efficiency map (**a**) ICE efficiency map; (**b**) MG1 efficiency map; (**c**) MG2 efficiency map.

*5.2. Simulation Result Analyses*

A comprehensive Matlab/Simulink model including driver, vehicle dynamics, eCVT DHT, ICE, MG1 and MG2, battery and control strategy is developed to demonstrate the feasibility of the proposed eCVT-based powertrain. Two driving cycles including a practical Chinese city cycle and the well-known Worldwide harmonized Light vehicles Test Cycle (WLTC) are employed in simulation.

A control strategy based on predetermined optimal control profile of ICE torque by rule of thumb is employed to investigate the fuel-saving feasibility of the proposed eCVT hybrid powertrain. Further optimization on control strategy based on online efficiency optimization of the power sources of ICE, MG1 and MG2 will be investigated in the future. However, even the current simple control strategy can fully demonstrate that significant fuel-consumption improvement to the hybrid pickup truck can be achieved by the proposed eCVT with two gears.

5.2.1. Simulation Investigation for an Actual Chinese Drive Cycle

Figure 12 presents the simulation results for three cycles of a measured city drive cycle in the Baoding city of China. The eCVT DHT control mode shown in Figure 12a is defined as 0 = EV mode, 1 = eCVT1 and 2 = eCVT2 mode, PH1 and PH2 are included, respectively within eCVT1 and eCVT2 as their special cases. The eCVT powertrain system intermittently operates in EV, eCVT1 and eCVT2 mode. During urban driving, the EV mode is automatically applied whenever the battery SOC reaches above the predefined threshold of 40% and battery charge mode is not requested. The eCVT2 hybrid mode is primarily applied to satisfy the light-to-medium driver torque request and charging electric power to the onboard traction battery through MG2 power splitting. The battery shall request a

charging mode when the battery SOC drops below 30% in an EV mode. The battery charging mode request within an eCVT mode shall be deactivated once the battery SOC exceeds 55%. Based on Figure 12d, the HV battery SOC is maintained within its predefined operational window of 30%–70%. The eCVT1 hybrid mode is only applied occasionally whenever the driver load request exceeds the eCVT2 traction capability.

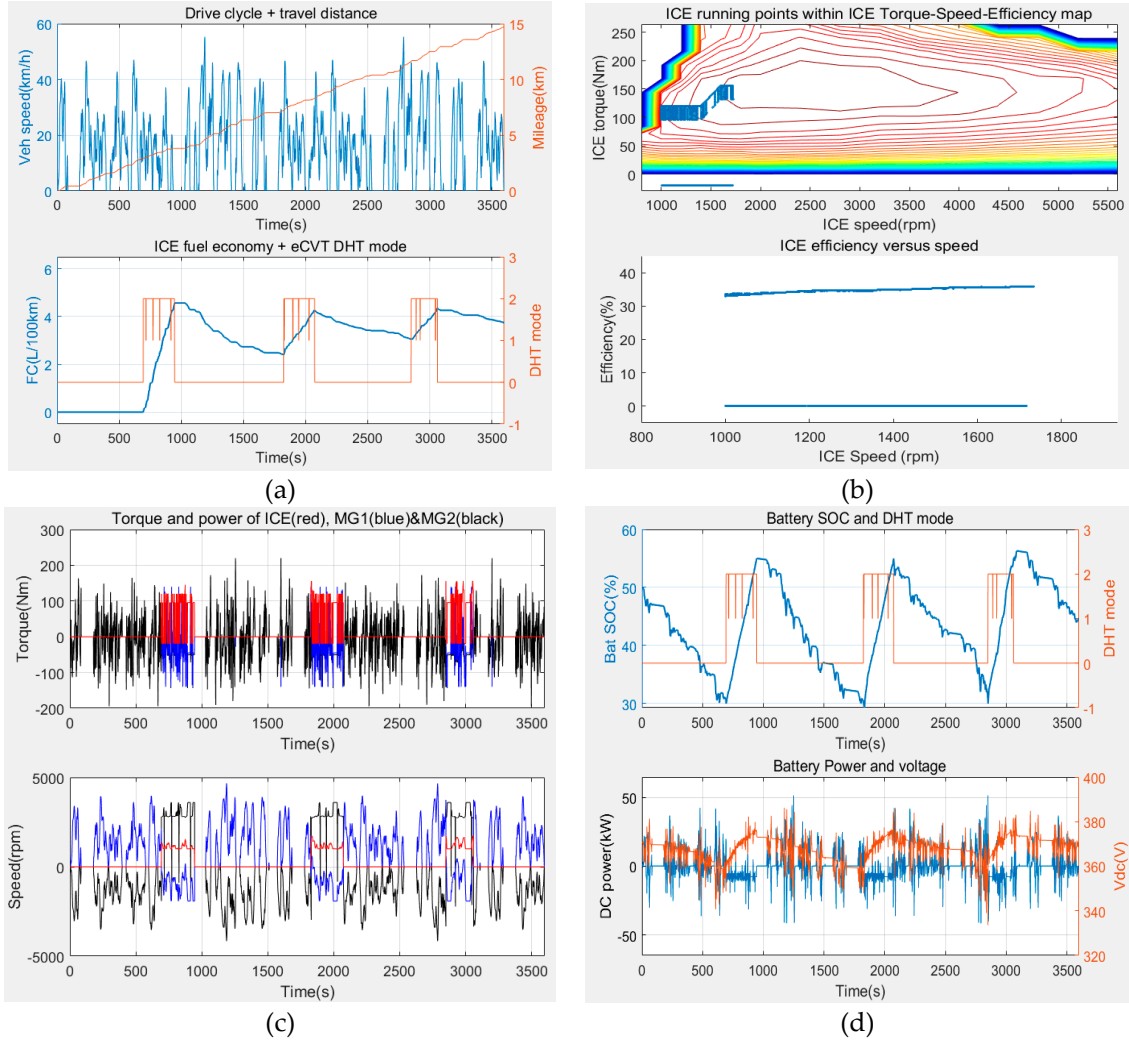

**Figure 12.** Simulation results for three Baoding city driving cycles: (**a**) Baoding city cycle data, DHT mode and ICE fuel consumption; (**b**) ICE operational torque location within ICE map and efficiency; (**c**) ICE/MG1/MG2 operational torque and speed; (**d**) DHT mode, high-voltage (HV) battery state of charge (SOC), power and voltage.

For this city driving cycle with many stops at traffic lights, the EV mode dominates most of the driving time as indicated in Figure 12c. ICE is only started when battery charging is requested. In the eCVT modes, ICE provides powertrain traction torque control and in the meantime delivers extra power to charge the HV battery through power splitting by either MG1 or MG2. Figure 12b indicates that ICE is controlled to operate within a narrow band in blue of the pre-determined optimal torque-speed curve specified in Figure 11a. As a result, the overall ICE efficiency is over 30%. Inefficient low ICE torque requests are prohibited by the control strategy. The negative ICE line in Figure 12b is due to friction drag torques during ICE fuel cut-off operation if ICE rotates above 0 rpm. The instant ICE fuel consumption (FC) illustrated in Figure 12a is calculated based on the integral value of ICE fuel in liter being divided by integrated travel distance in 100 km at every sampling interval. An average

ICE fuel consumption of less than 4 L/100 km can be achieved for this light-load Baoding city-driving cycle if a balanced battery SOC is considered.

Figure 12c illustrates the operational behavior of three power control sources of ICE in red, MG1 in blue and MG2 in black. In the EV mode, MG2 is fully in charge of the torque and power delivery for satisfying the drive demands; ICE keeps at standstill with 0 Nm torque output; MG1 rotates due to the PG1 mechanical restriction without active torque and power output. During the forced battery charging and SOC-balancing control by eCVT2, MG2 runs as speeder to implement torque and power split function to ICE and MG1 inputs. ICE is always optimized to deliver efficient torque through its mechanical pass to meet the driver demand. In the meantime, MG1 assists to provide instant acceleration torque or regenerative braking torque occasionally.

The battery SOC shown in Figure 12d is balanced within its permissible operation window of 30%–70%. The battery SOC is integrated based on the battery load current due to the electric loads of MG1, MG2 and low-voltage (LV) 12 V with a 500 W constant LV load assumed in simulation. Due to the HV battery internal impedance and SOC variation impact, the battery voltage changes significantly with electric current load. The battery power is clamped within its permissible power range within ±60 kW.

### 5.2.2. Simulation Investigation for A Complete WLTC Cycle

The simulation demonstration results for three complete WLTC cycles are illustrated in Figure 13. The overall average fuel consumption reaches about 6 L/100 km with respect to a balanced battery SOC. The EV mode is still used frequently during city driving, and the eCVT2 mode is primarily adopted for battery SOC balancing and fuel-efficient driving. The parallel hybrid mode PH2 in eCVT2 is introduced for direct engine drive at MP2 during highway driving while MG2 is maintained at 0rpm, as indicated in the highlighted band of Figure 13c. ICE is in many cases maintained to follow the predefined optimal performance curve but has more PH2 parallel operation to support highway driving in more efficient way. The overall ICE efficiency is kept above 30%. The battery SOC is controlled within 30%–70% and the battery power is limited within its power boundary of −60–60 kW.

### 5.2.3. Acceleration Performance Investigation

A full acceleration driver request is introduced to demonstrate the acceleration performance of the pickup HEV vehicle, as illustrated in Figure 14. eCVT1 is primarily selected to achieve fastest acceleration from the vehicle standstill up to the vehicle speed of 105 km/h. The maximum traction torque from both ICE and MG2 applied simultaneously at the beginning and MG2 torque gradually decreases due to overall eCVT power restriction. MG1 operates in a closed-loop feedback speed control to deliver torque and power split to ICE input during the eCVT1 mode. Once the vehicle speed exceeds the eCVT1 speed limit of 105 km/h, ECVT2 is employed instead to keep high-speed driving. ICE still maintains its maximum torque output and MG1 provides assistant torque for maintaining full acceleration instead. Alternatively, MG2 is swapped into closed-loop speed control in order to automatically split the torque and power input from both ICE and MG1. A final acceleration performance with about 9.7 s from 0 to 100 km/h and 4.4 s from 0 to 60 km/h is achieved.

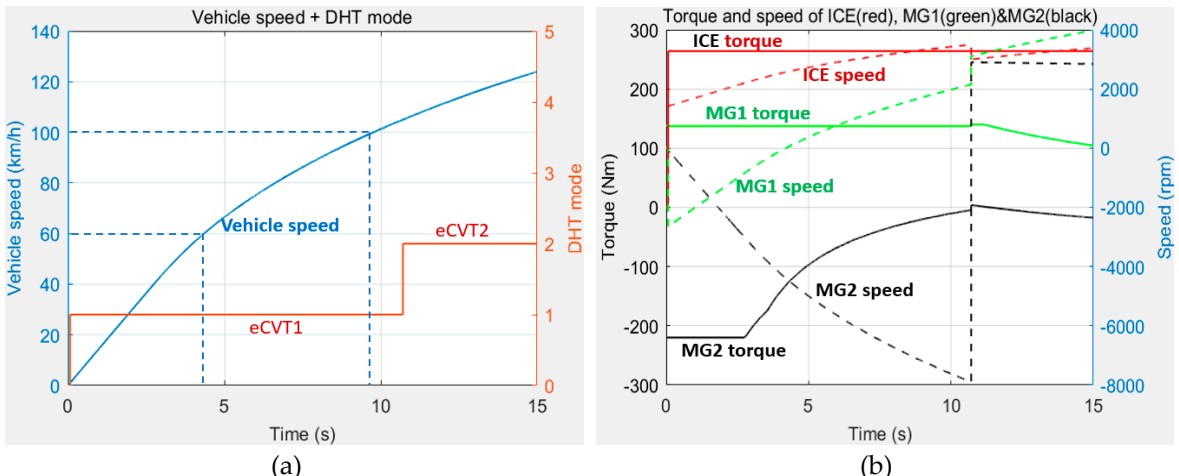

**Figure 13.** Simulation results for three Worldwide harmonized Light vehicles Test Cycle (WLTC) cycles: (**a**) WLTC cycle data, DHT mode and ICE fuel consumption; (**b**) ICE operational torque location within ICE map and efficiency; (**c**) ICE/MG1/MG2 operational torque and speed; (**d**) DHT mode, HV battery SOC, power and voltage.

**Figure 14.** Acceleration simulation results with a full-pedal driver request: (**a**) vehicle speed and driving mode; (**b**) ICE/MG1/MG2 torque and speed.

## 6. Conclusions

A novel two-mode power-split eCVT hybrid transmission architecture with two planetary gears for HEV/PHEV applications is proposed. The actuation of one clutch and one brake inside eCVT is controlled for implementing multimode hybridization and EV functions. The detailed theoretical analyses illustrate that the proposed eCVT can offer many advantages over several popular hybrid powertrains already available in market. The direct mechanical path of engine torque with two independent gear ratios can provide a maximum ICE torque output available from standstill with two different gear ratios. Two electric machines can be sized with much less torque and power rating due to the availability of maximum engine torque transmitted mechanically by separate two gear ratios from standstill. The instantaneous torque capability of two-geared eCVT is comparable to a conventional multi-geared transmission powertrain system. Comprehensive simulation demonstrations for a pickup truck HEV application are carried out to prove that significant fuel saving and acceleration performance improvement can be achieved.

## 7. Patents

The proposed eCVT DHT architecture concept in this paper is fully based on the Chinese patents of CN201921174082.6 and CN201910672384.4.

**Author Contributions:** Conceptualization, X.C. (Xiaojiang Chen) and J.J.; methodology, X.C. and L.Z.; software, X.C. (Xiaojiang Chen); validation, X.C. (Xiaojiang Chen), H.T. and J.J.; formal analysis, X.C. (Xiaojiang Chen); investigation, J.J.; resources, L.Z., and X.C.(Xiaofeng Chen); data curation, H.T.; writing—original draft preparation, X.C. (Xiaojiang Chen); writing—review and editing, J.J., L.Z., and X.C. (Xiaofeng Chen); visualization, X.C. (Xiaojiang Chen).; supervision, L.Z.; project administration, J.J., and X.C. (Xiaofeng Chen); funding acquisition, H.T. All authors have read and agreed to the published version of the manuscript.

**Funding:** This research received no external funding.

**Conflicts of Interest:** The authors declare no conflict of interest.

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
