# Peer review of "Study and Analysis of a Multi-Mode Power Split Hybrid Transmission"

_wevj, doi:10.3390/wevj11020046_

Round 1
Reviewer 1 Report
As a general comment, the reviewer would recommend the authors to take advantage of an English native speaker for the publication. Throughout the article the grammar, the spelling together with the presence of short periods without any linking words make the overall presentation hard to follow and to be understood. Therefore, the reviewer would recommend to re-write the article in toto.
Major and deep revisions have to be made in order to publish this article. More specifically, the reviewer would recommend to simulate the pickup HEV in more than just two cycles and to use real world drive data in order to provide more consistency to the results obtained. Besides, the control strategy used for simulating the HEV should be carefully chosen and described to ensure the reliability of the fuel consumption values produced.
A more detailed transcript of the major issues is hereafter reported for the authors’ convenience.
- Section 1: Introduction
The reviewer would recommend to make examples of the “more stringent emission regulation” that the authors refer to, as well as to explain more clearly which “diesel emission scandal” the authors mention. By doing so, the reader could understand easily what the authors are talking about.
The authors mention the “lack of electric hybrid technologies” and then refer to a P2 PHEV, isn’t it part of what the authors say is missing in Europe? Please clarify this point.
Lines 38-40 and 43-46. The authors make two important statements regarding the fuel efficiency and costs of HEV and then P2 HEV. Would the authors consider to cite where these thoughts come from?
Lines 47-50. Would the authors consider to clarify who proposes the architecture mentioned in these lines?
Lines 54-66. Would the authors consider to use a bullet list in order to explain in a clearer way what they are saying? If not possible, then the reviewer would recommend to use linking words to smoother the reading of these lines.
Line 67. The authors refer to “serial” hybrid, the reviewer would recommend to substitute this word with series.
Lines 70-77. The reviewer would recommend to link the several sentences by using connecting words, the presence of all these short phrases do not help the reader in understanding what the authors are saying.
Moreover, throughout the whole chapter there are several typos, such as “power-slip” in line 92 and “power-spilt” in line 97. The reviewer would recommend also to pay more attention in the time used for the verbs.
- Section 3: Theoretical Analysis on Drive Modes
Line 161. What do the authors mean when referring to “whenever the traction battery state of charge (SOC) is allowed”?
Lines 212-216, Lines 239-243, Lines 255-257. The reviewer would suggest to rephrase the sentences in order to make the explanation clearer and let the reader understand in a deeper way what the authors are saying.
The reviewer would recommend the usage of synonyms instead of repeating words such as “therefore”, “ignorable” and so on that can be found in the whole chapter.
- Section 4: Component sizing Principals
How do the authors selected the values of and ? Were they given or did the authors make computations in order to achieve these numbers?
The authors refer more than once to “axel”, the reviewer would suggest to use the correct noun “axle” instead.
Figure 10. The reviewer would recommend to use a contour map for the graphs just mentioned since it would help the reader to understand the efficiency maps presented by the authors. Moreover, the authors use as dimension of figure 10-a “km/h” when referring to ICE speed which is clearly a mistake. Besides, the reviewer would recommend to show the colormap in these figures so to enable a easier reading of the efficiency maps.
- Section 5: Simulation Results Analyses
Would the authors explain the usage of a section of the WLTP cycle in order to represent an urban drive profile? Would have it been better to refer to a different cycle instead of repeating the first two sections of the WLTP?
Why do the authors choose to simulate the pickup HEV only in these two cycles? What about using real world driving data and/or other standard drive cycles?
Line 434-436. Would the authors please clarify the “rule of thumb” used for controlling the HEV? Why did the authors not use one of the algorithm worldwide spread for controlling the HEV, such as Dynamic Programming (DP)?
Lines 446-447. The authors refer to “whenever the battery SOC status is permitted”. Would the authors please clarify what they mean with this sentence?
Lines 478-483. How did the authors define the SOC window used for the simulations? The reviewer would recommend to decrease the SOC window (currently 30% - 70%) in order to consider also the battery state of health which would be threatened by the continuous charging/discharging phases that can be seen in the simulations.
Figure 11 and Figure 12. The reviewer would recommend to make changes to these two charts since they are hard to understand. Firstly, the reviewer would suggest to show the instantaneous fuel consumption in g/s instead of the fuel consumption in L/100 km, which is related to the overall cycle. Besides, when referring to the ICE efficiency the reviewer would recommend to omit the zero efficiency points that presumably are obtained when the ICE is disactivated.
In both the case studies, the authors show that the initial SOC values is of about 50%, whereas the final one varies. In the first case is of 55% ca. whereas the second one ends with a SOC of about 65%. How did the authors constrain, if they did, the final value? Do the authors refer to any regulations when testing the fuel consumption of the HEV? The reviewer would recommend to read the SAE J1711 or to consider to have almost the same value for initial and final SOC (difference of few percentages points are acceptable), in order to present a fuel consumption which does not depend on the battery usage. The reviewer would suggest to include these constraints when controlling the HEV.
Lines 488-490. The authors mention the “direct engine drive” and refers to figure 12-c to explain it. The reviewer would recommend to zoom in the graphs in order to show to the reader what the authors are saying and make it more clear to understand.
Why do the authors show an analysis of the acceleration from 0 to 120 km/h ca. when the performances later on mentioned are the 0-100 km/h and the 0-60 km/h times?
Reviewer 2 Report
This paper proposes a novel two-mode power-split eCVT hybrid transmission architecture with two planetary gears for HEV/PHEV applications. In addition, detailed theoretical analyses of the power split modes are carried out. The simulation demonstrations of pickup truck hybrid vehicle application also proved the system's ability to save fuel and improve acceleration performance.
For the content of the paper, there are the following problems:
1、Please enrich the literate in the introduction section and carefully check the grammar mistakes in the paper.
2、The Toyota HSD and Honda iMMD in the article are single-row planetary gears. For the two planetary gears architecture, the current research also includes configurations such as Ravigneaux planetary system and THS-II generation. Can you consider comparing the design scheme of this article with them?
3、The structure type designed in this paper changes the radial size and symmetry, please briefly discuss the possibility of the layout type in product realization.
Round 2
Reviewer 1 Report
Please, consider about further revising the language
Reviewer 2 Report
The reviewer satisfied with the latest version.